# Bayesian spatiotemporal inference of trace gas emissions using an integrated nested Laplacian approximation and Gaussian Markov random fields

Luke M. Western[1], Zhe Sha[2,3], Matthew Rigby[1], Anita L. Ganesan[2], Alistair J. Manning[4], Kieran M. Stanley[1,5], Simon J. O'Doherty[1], Dickon Young[1], and Jonathan Rougier[3]

[1]School of Chemistry, University of Bristol, Bristol, UK
[2]School of Geographical Sciences, University of Bristol, Bristol, UK
[3]School of Mathematics, University of Bristol, Bristol, UK
[4]Hadley Centre, Met Office, Exeter, UK
[5]Institute for Atmospheric and Environmental Science, Goethe University Frankfurt, Frankfurt am Main, Germany

**Correspondence:** Luke M. Western (luke.western@bristol.ac.uk)

**Abstract.** We present a method to infer spatially and spatiotemporally correlated emissions of greenhouse gases from atmospheric measurements and a chemical transport model. The method allows fast computation of spatial emissions using a hierarchical Bayesian framework as an alternative to Markov chain Monte Carlo algorithms. The spatial emissions follow a Gaussian process with a Matérn correlation structure which can be represented by a Gaussian Markov random field through a stochastic partial differential equation approach. The inference is based on an integrated nested Laplacian approximation (INLA) for hierarchical models with Gaussian latent fields. Combining an autoregressive temporal correlation and the Matérn field provides a full spatiotemporal correlation structure. We first demonstrate the method on a synthetic data example and follow this using a well-studied test case of inferring UK methane emissions from tall tower measurements of atmospheric mole fraction. Results from these two test cases show that this method can accurately estimate regional greenhouse gas emissions, accounting for spatiotemporal uncertainties that have traditionally been neglected in atmospheric inverse modelling.

## 1 Introduction

Emissions of greenhouse gases, ozone depleting substances and air pollutants are increasingly inferred indirectly from atmospheric trace gas concentration observations and chemical transport models. These "top-down" or "inverse" methods complement inventory- or process-model-based "bottom-up" techniques that are used, for example, in national reporting of greenhouse gas emissions to the United Nations Framework Convention on Climate Change (UNFCCC, e.g. Leip et al., 2018).

Top-down methods rely on some form of statistical inference, or inverse theory, to infer emissions at global (e.g. Saunois et al., 2016) to regional (e.g. Brunner et al., 2017) scales. They also require a chemical transport model to provide the relationship between atmospheric mole fraction and emissions. The most common type of inverse method uses Bayesian inference. Bayesian inference relies on the information about the uncertainties in the measurement, transport model and prior probability on the parameters to inform the emissions estimate. Inverse methods often assume that uncertainties in the likelihood and prior

probabilities are known exactly and Gaussian (e.g. Stohl et al., 2009; Brioude et al., 2013). These assumptions allow large inverse problems to be solved efficiently. However, when uncertainties are poorly understood, Bayesian methods have been shown to lead posterior solutions that are highly dependent on these assumptions (e.g. Ganesan et al., 2014).

Recently, hierarchical Bayesian schemes have been developed to infer unknown uncertainties in the inversion framework
(e.g. Ganesan et al., 2014; Jeong et al., 2016; Lunt et al., 2016). These hierarchical Bayesian inference schemes use Markov chain Monte Carlo (MCMC) methods that are well-suited to small-dimensional problems. However, they can suffer from issues with convergence, especially as the dimension of the problem increases (Cowles and Carlin, 1996). Hierarchical inference of spatially and spatiotemporally correlated emissions is computationally difficult and currently relies on methods such as Kalman filters with assumed covariance structures (e.g. Brunner et al., 2012) or an empirical hierarchical framework, where unknown
hyperparameters are not integrated out during inference (e.g. Michalak et al., 2005; Berchet et al., 2015). Previous work shows, however, that including spatial correlation improves the fit between modelled and observed data (Zammit-Mangion et al., 2016), while temporal correlation is important to represent the dependence of emissions between time periods. The size and spatial coverage of measurement networks available for inferring gas emissions is growing, particularly through satellite observations (e.g. Bergamaschi et al., 2007; Veefkind et al., 2012; Wecht et al., 2014; Ganesan et al., 2017). Therefore, there is a need to
develop methods that can utilise these big data sets, whilst maintaining the benefits of uncertainty quantification in hierarchical methods, ideally extending to spatiotemporal inference.

This work presents a computationally efficient hierarchical Bayesian framework for inferring spatiotemporally correlated trace gas emissions in a widely-used regional atmospheric chemical transport modelling framework. We use an integrated nested Laplacian approximation (INLA) for the Bayesian inference. The spatial correlation structure with spatial Markov
properties results from the Gaussian random field being a solution a particular stochastic partial differential equation. Kronecker product algebra allows efficient extension to spatiotemporal correlation. Section 2 presents the construction of the hierarchical problem, describes the formation of the correlation structure and introduces INLA. Section 3 contains two case studies as a proof-of-concept for the method with a discussion of their implementation. The first is an example using four consecutive periods of pseudo-data observations of methane from four UK monitoring stations. This then extends to using real observations
from the four measurement stations to infer UK emissions for 2014, split into four three monthly periods. Section 4 discusses the results and computational performance of the method and section 5 presents the conclusions of the study.

## 2 Methods

This section details an efficient approach to forming spatial and spatiotemporal correlation functions and outlines how this applies to the inference of regional trace gas emissions from measurements using fast inference for hierarchical models. We
limit the scope of this paper to the well-established problem of regional inference of long-lived trace gas emissions using a backward-running Lagrangian particle dispersion model (Stohl et al., 2009; Manning et al., 2011; Henne et al., 2016). In this framework, the model directly calculates the sensitivity of the measurements to emissions from each grid cell in the domain. Extension to other systems (e.g. global models) would require modification to provide sensitivity on a global scale with

substantially different temporal and spatial emissions. We begin by introducing the model and the inferred latent parameters (the emission fluxes and boundary conditions), followed by an introduction to Gaussian Markov random fields and how they are useful for efficient calculation of spatial and spatiotemporal correlation structures. All together, this forms a Bayesian hierarchical model, from which emissions can be inferred.

## 2.1 Model framework

The aim is to infer some parameters of interest, here a spatial field of a priori emissions scaled by some factor, $\mathbf{x}$, from some measurement. The a priori emissions at given location are generally informed by spatially resolved bottom-up inventories (as in section 3.1.3) or extrapolation from some reported emissions value. The value $\mathbf{x}$ is a multiplicative scaling of this a priori value for emissions, most generally expressed as a quantity of gas per unit time per unit area. The approach taken in this work uses a Gaussian Markov random field for fast and efficient calculation of spatial correlation for $\mathbf{x}$. This means that the emissions field is required to be a latent Gaussian field, which will be discussed further in section 2.2. Net surface fluxes of many greenhouse gases are positive, at the scales resolved by the model. In this work, due to the usage of a latent Gaussian field, which must be defined over both positive and negative values, we choose to look at the deviations of emissions from the prior mean emissions field. This is in an effort to fit the physical model to the imposed statistical model that fast computation requires. Taking the approach that, for many regional inverse problems involving longer-lived trace gases, there is a linear relationship between emissions which are constant in time and observed atmospheric concentration, the relationship between measurements and emissions is,

$$\mathbf{y} = \mathbf{H}\mathbf{x} + \mathbf{K}\mathbf{u} + \epsilon, \tag{1}$$

where $\mathbf{y}$ is a vector of the residual between the measured and a priori predicted measurement, $\mathbf{H}$ is a Jacobian (or sensitivity) matrix, which maps the surface emissions to the measurements, $\mathbf{u}$ is a vector of independent and identically distributed variables containing the contribution to the measurement of mole fractions at the boundary of the domain minus the prior mean contribution, with an associated sensitivity matrix $\mathbf{K}$, and $\epsilon$ is some stochastic error. The variable $\mathbf{x}$ has a Gaussian prior probability with zero expectation and a covariance described by a Gaussian random field. This will be solved using a hierarchical framework (section 2.4).

## 2.2 Gaussian Markov random fields

The emissions scaling from its a priori value $\mathbf{x}$ is spatially continuous over the domain of interest. We assume that it exhibits a spatial correlation structure, because emissions at one location are generally not independent from all other locations in the same field. We choose to model the covariance in this field using a Matérn covariance function, which Stein (1999) shows is well suited to natural systems due to its flexibility, with other correlation structures (e.g. exponential) being special cases of the Matérn family (Guttorp and Gneiting, 2006). A stationary Gaussian random field with a Matérn covariance is the solution to a particular stochastic partial differential equation (Whittle, 1954, 1963), given by

$$(\kappa^2 - \Delta)^{\alpha/2}(\tau x(\mathbf{s})) = \mathcal{W}(\mathbf{s}), \quad \mathbf{s} \in \Omega, \tag{2}$$

where $\kappa$ is the spatial scale parameter, $\tau$ influences the variance, $\Delta$ is the Laplacian operator, $\Omega$ is the spatial domain and $\mathcal{W}(\mathbf{s})$ is a Gaussian stochastic process for locations $\mathbf{s}$, noting that the dependence is only on the Euclidean distance and so the process is isotropic. For an example of non-stationary and anisotropic fields see e.g. Marques et al. (2019). The smoothness parameter $\alpha$ gives a continuous domain Markov field for integer values and is set to $\alpha = 2$ (see Whittle, 1954). Smaller values will give more short-scale variability and can be difficult to differentiate from noise. Lindgren et al. (2011) show that if the field is represented using a Gaussian Markov random field (see Rue and Held, 2005) then it is very efficient to directly construct the precision, or inverse covariance, matrix using a basis function representation

$$x(\mathbf{s}) = \sum_k \psi_k(\mathbf{s}) x_k, \tag{3}$$

where $\psi_k(\cdot)$ is a piecewise linear function that in each triangle of a mesh construction of the spatial domain, where $\psi_k(\cdot)$ is 1 at vertex $k$ and 0 at all other vertices. This mesh is created using constrained refined Delaunay triangulation (Shewchuk, 2002), placing nodes at the main points of interests and infilling the rest of the space using some condition of minimum and maximum length of the vertices. In this work we choose to represent the UK and Ireland with an evenly spaced denser mesh with a coarser mesh outside of this region. A mesh could be further refined, for example by creating an even denser mesh close to the measurement site where sensitivity is higher. It is important to extend the mesh beyond the region where the measurements are sensitive to emissions as the mesh is constructed using Neumann boundary conditions, which trigger reflection and therefore overestimation close to the boundaries. This has no effect on the inferred emissions, provided that the mesh is extended far enough around the domain of interest. Figure 1 shows the mesh used in this work, excluding the extended outer mesh region.

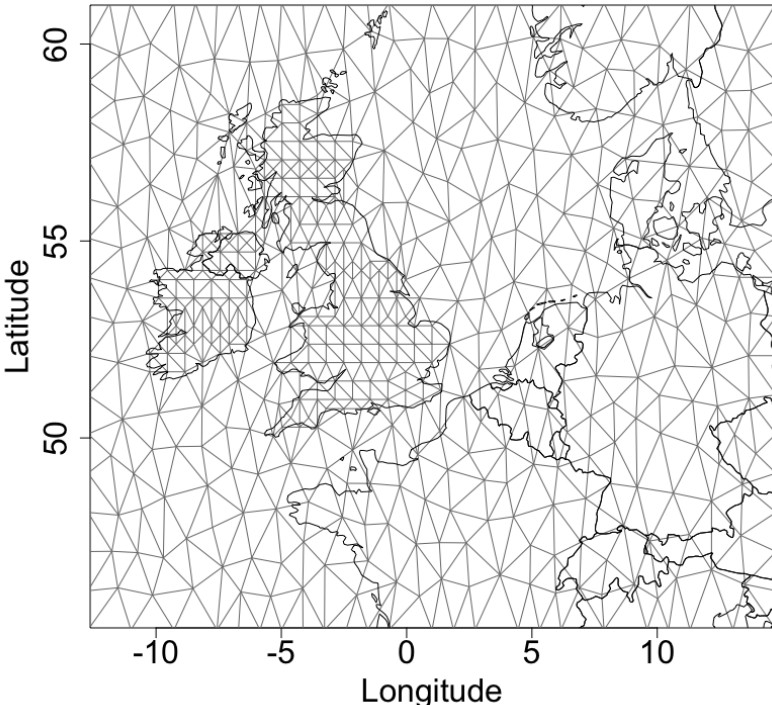

**Figure 1.** A mesh constructed using constrained refined Delaunay triangulation, where the distribution of nodes is denser around the United Kingdom and Ireland.

## 2.3 Extension to spatiotemporal correlation

A spatiotemporal extension to the forward model 1 is possible by including the spatial correlation structure introduced in section 2.2 in a temporal framework (Cameletti et al., 2013). Similar to equation 1, the deviation from the prior mean measurement at site $l$ made at time $t$ in its simplest form is

$$y_{tl} = \sum_{i=1}^{n} h_{itl} x_{it} + \sum_{j=1}^{4} k_{jtl} u_{jt} + \epsilon_{tl}, \tag{4}$$

where $i$ represents each of the total $n$ nodes and $j$ represents the edge of the domain at each of the four cardinal directions. We make the assumption that measurements made at a given time are independent, giving a vectorised observation vector $\mathbf{y}_t$ at

each time. This can be generalised to include correlated measurements, where the covariance function should result in a sparse matrix to retain computational efficiency. Following this, the matrix $\mathbf{H}$ from (1) becomes the sparse block diagonal matrix

$$\mathcal{H} = \begin{bmatrix} \mathbf{H}_1 & 0 & 0 & \dots & 0 \\ 0 & \mathbf{H}_2 & 0 & \dots & 0 \\ \vdots & \vdots & \vdots & \ddots & \\ 0 & 0 & 0 & & \mathbf{H}_m \end{bmatrix}, \tag{5}$$

which operates on the vectorised spatiotemporal scaling of the emissions field $\mathbf{x} = [\mathbf{x}_1, \mathbf{x}_2, \dots \mathbf{x}_m]$ to model the observations $\mathbf{y} = [\mathbf{y}_1, \mathbf{y}_2, \dots \mathbf{y}_m]$. The time varying structure of $\mathbf{H}_t$ and $\mathbf{x}$ applies also to $\mathbf{K}_t$, and $\mathbf{u}_t$. We impose the temporal correlation structure between emissions over time to be an autoregressive model of order one, where

$$\mathbf{x}_t = \phi \mathbf{x}_{t-1} + \delta_t; \quad \delta_t \sim \mathcal{N}\left(0, \mathbf{Q}_S^{-1}\right) \quad t = 1 \dots m, \tag{6a}$$

and

$$\mathbf{x}_1 \sim \mathcal{N}\left(0, \left(\frac{\mathbf{Q}_S}{1-\phi^2}\right)^{-1}\right), \tag{6b}$$

where $\phi$ is the temporal correlation and $\mathbf{Q}_S^{-1}$ is the spatial correlation structure described by a Matérn field using the stochastic partial differential equation approach for a Gaussian Markov random field. We have chosen an autoregressive model of order one as we believe that emissions are generally similar to those at previous time step. In the given model, $\phi$ is the correlation between the previous time step and the current time step. This means that the emissions at time $t$ have a similarity of $\phi$ to emissions at $t-1$ plus some spatially correlated random effect $\delta_t$. The vectorisation of $\mathbf{x}$ allows a separable covariance structure for the temporal and spatial covariances, which means that the spatiotemporal precision matrix can be expressed using a Kronecker product as follows (Mardia et al., 1979),

$$\mathbf{Q} = \mathbf{Q}_T \otimes \mathbf{Q}_S. \tag{7}$$

Estimating hourly emissions at each time $t$ soon makes inference prohibitive due to burden. Instead we make the assumption that emissions are constant over a three-month period to reduce the computational size. We continue with the assumption that measurements within a single time period are independent, although this can be generalised if required. In this experiment we make the assumption that emissions are constant over a three-month period and that the correlation between these three monthly periods is autoregressive of order one.

## 2.4 Hierarchical model

Inferring the emissions and the related uncertainties requires a hierarchical model to infer the quantities of interest from measurements, while estimating some unknown parameters which are necessary for inference. The main focus of this work is to estimate the posterior distribution of the emissions field $\mathbf{x}$, based on observations $\mathbf{y}$. We follow a typical Bayesian hierarchical

framework

$$p(\mathbf{x}, \mathbf{u}, \boldsymbol{\theta} \mid \mathbf{y}) \propto p(\mathbf{y} \mid \mathbf{x}, \mathbf{u}, \boldsymbol{\theta}) \, p(\mathbf{x} \mid \mathbf{u}, \boldsymbol{\theta}) \, p(\mathbf{u} \mid \boldsymbol{\theta}) \, p(\boldsymbol{\theta}) \tag{8}$$

where $\boldsymbol{\theta}$ is vector of hyperparameters describing the variances and covariances in $\mathbf{x}$, $\mathbf{u}$ and $\mathbf{y}$, noting that $\mathbf{x} \perp\!\!\!\perp \mathbf{y} \mid \boldsymbol{\theta}$. We assume

$$\begin{aligned}
\mathbf{y} \mid \mathbf{x}, \mathbf{u}, \boldsymbol{\theta} &\sim \mathcal{N}(\mathcal{H}\mathbf{x} + \mathbf{K}\mathbf{u}, \mathbf{Q_y}(\boldsymbol{\theta})^{-1}), \\
\mathbf{x} \mid \mathbf{u}, \boldsymbol{\theta} &\sim \mathcal{N}(0, \mathbf{Q}(\boldsymbol{\theta})^{-1}), \\
\mathbf{u} \mid \boldsymbol{\theta} &\sim \mathcal{N}(0, \mathbf{I}\sigma_{\mathrm{BC}}^2), \\
\boldsymbol{\theta} &\sim p(\boldsymbol{\theta}).
\end{aligned}$$

where the precision matrix of the model-measurement uncertainty $\mathbf{Q_y}$ contains the hyperparameter for the standard deviation of the model-measurement standard deviation $\sigma_y$, $\sigma_{\mathrm{BC}}$ is the standard deviation of the prior for $\mathbf{u}$ and $\mathbf{I}$ is the identity matrix. Together the hyperparameters make the vector $\boldsymbol{\theta} = (\rho, \sigma, \phi, \sigma_{\mathrm{BC}}, \sigma_y)$, which have independent prior distributions. The hyperparameters for the spatial precision matrix $\mathbf{Q}(\boldsymbol{\theta})$ are transformations of the variables in (2),

$$\rho = \frac{\sqrt{8}}{\kappa}, \tag{10a}$$

$$\sigma = \left( \frac{1}{\sqrt{4\pi}\kappa\tau} \right)^{-\frac{1}{2}}, \tag{10b}$$

where $\rho$ is the range parameter and $\sigma$ is the marginal standard deviation of the latent field (Lindgren et al., 2011). We use penalised complexity priors to define the prior probabilities for these parameters (Simpson et al., 2017; Fuglstad et al., 2018). Penalised complexity priors allow the formation of priors when there is only a vague understanding of their true values, while enforcing more constraint than using a broad uniform or a Jeffreys prior. This uses the information loss of deviating from some baseline estimate of the parameter. Penalised priors do not increase the computational speed for the given case and are chosen instead for their intuitiveness. Penalised complexity priors are specified by the probability of the parameters being less than or greater than some baseline value,

$$p(\rho > \rho_0) = p_\rho,$$
$$p(\sigma < \sigma_0) = p_\sigma,$$

where $\rho_0$ and $\sigma_0$ are the baseline values and $p_\rho$ and $p_\sigma$ are the associated probabilities defined by the user. The hyperparameter $\phi$ controls the temporal correlation between latent variables and its prior probability is defined on a beta distribution scaled between minus one and one as

$$\phi \sim \mathrm{Beta}(a, b),$$

where $a$ and $b$ are assumed coefficients. The matrix $\mathbf{Q_y}$ is the precision matrix for the combined measurement and model errors. The diagonal of $\mathbf{Q_y}$ contains the square of the hyperparameter $\sigma_y$, where the prior probability follows defined on $\log \frac{1}{\sigma_y^2}$, is

$$\log \frac{1}{\sigma_y^2} \sim \mathcal{N}(\mu_{\sigma_y}, \sigma_{\sigma_y}^2)$$

The marginal standard deviation of the a priori boundary conditions $\sigma_{\mathrm{BC}}$ are also constructed this way, giving

$$\log \frac{1}{\sigma_{\mathrm{BC}}^2} \sim \mathcal{N}(\mu_{\sigma_{\mathrm{BC}}}, \sigma_{\sigma_{\mathrm{BC}}}^2).$$

A Bayesian hierarchical model requires a method of inference to estimate the parameters of interest and any parameters that are not of direct interest but required, and uncertain, in order to infer the other parameters in the hierarchy. Many methods of inference exist and have been applied to the problem of estimating emissions of trace gases (see references in section 1). A promising approach using the correlation structure in section 2.2 and 2.3 is INLA (Lindgren and Rue, 2015). Section 2.5
outlines this approach to inference.

## 2.5   Inference using an integrated nested Laplacian approximation

An integrated nested Laplacian approximation (Rue et al., 2009) provides a fast and efficient framework to infer the latent variable $\mathbf{x}$ and hyperparameters $\boldsymbol{\theta}$ from measurements $\mathbf{y}$. The calculation of the INLA is possible using the R-INLA package (Lindgren and Rue, 2015). In this work we use R-INLA version 17.06.20. The speed in this approach comes from solving the
marginal posteriors for $x_i$, i.e. each element $i$ of the latent field, through the numerical integration

$$p(\theta_j \mid \mathbf{y}) = \int p(\boldsymbol{\theta} \mid \mathbf{y}) \mathrm{d}\boldsymbol{\theta}_{-j}, \tag{12}$$

$$p(x_i \mid \mathbf{y}) = \int p(x_i \mid \mathbf{y}, \boldsymbol{\theta}) \, p(\boldsymbol{\theta} \mid \mathbf{y}) \mathrm{d}\boldsymbol{\theta}. \tag{13}$$

where $j$ is the $j$th element in $\boldsymbol{\theta}$ and $-j$ indicates all by the element $j$. Equations 12 and 13 make use of a Laplace approximation,
by approximating $p(\boldsymbol{\theta} \mid \mathbf{y})$ by

$$p(\boldsymbol{\theta} \mid \mathbf{y}) \propto \left. \frac{p(\mathbf{x}, \boldsymbol{\theta}, \mathbf{y})}{p_G(\mathbf{x} \mid \boldsymbol{\theta}, \mathbf{y})} \right|_{\mathbf{x} = \mathbf{x}^*(\boldsymbol{\theta})}, \tag{14}$$

where $p_G(\mathbf{x} \mid \boldsymbol{\theta}, \mathbf{y})$ is a Gaussian approximation to the full conditional of $\mathbf{x}$ and the right hand is evaluated at $\mathbf{x}^*(\boldsymbol{\theta})$ which is the modal probability of $\mathbf{x}$ for a given $\boldsymbol{\theta}$. This approximation is exact if $p(\boldsymbol{\theta} \mid \mathbf{y})$ is Gaussian, and gives a good approximation for log concave problems (Tierney and Kadane, 1986). Then, it is possible to approximate $p(x_i \mid \mathbf{y}, \boldsymbol{\theta})$ using another Laplace
approximation

$$p_{\mathrm{LA}}(x_i \mid \boldsymbol{\theta}, \mathbf{y}) \propto \left. \frac{p(\mathbf{x}, \boldsymbol{\theta}, \mathbf{y})}{p_G(\mathbf{x}_{-i} \mid x_i, \boldsymbol{\theta}, \mathbf{y})} \right|_{\mathbf{x}_{-i} = \mathbf{x}_{-i}^*(x_i, \boldsymbol{\theta})}, \tag{15}$$

where $p_G(\mathbf{x}_{-i} \mid x_i, \boldsymbol{\theta}, \mathbf{y})$ is the Gaussian approximation to $\mathbf{x}_{-i} \mid x_i, \boldsymbol{\theta}, \mathbf{y}$ evaluated at the mode $\mathbf{x}^*_{-i}(x_i, \boldsymbol{\theta})$. See Rue et al. (2009) and Martins et al. (2013) for a more in depth description.

While this method relies on calculating only the marginal posterior distributions of $x_i$, it is still possible to predict a linear combination of the field to provide regional emissions totals (e.g. country totals). We define a linear predictor of emissions for a given region $\eta*$, defined using the basis function representation of the mesh where each node contains information on the spatial area represented by that node and its connecting vertices and zeros for all nodes that are not in the region of interest. Then we can approximate the linear combination of the parameters of interest, giving a combined emissions total $\eta*$, using equations 12 and 13 and transforming the predicted latent field by the weightings containing the area information.

## 3 Case studies

This section presents two case studies to demonstrate how the method applies to inferring trace gas emissions. The first uses simulated methane observations from four tall tower measurement sites to infer simulated spatiotemporal emissions from the UK. The second case study extends on the first case study by using real observations from the four tall towers to infer emissions of methane from the UK over four three monthly periods in 2014. While the size of this problem is not particularly large, we demonstrate the method using UK methane emissions as a proof of concept as it is a well studied test case (Manning et al., 2011; Ganesan et al., 2015; Lunt et al., 2015), which should exhibit a spatiotemporal correlation structure. The method can be extended to larger spatial domains or dataset sizes as required.

### 3.1 Transport model and data

#### 3.1.1 Measurement data

The case studies observations from four measurement sites, three in the UK and one in Ireland, which are part of the UK Deriving Emissions related to Climate Change (DECC) network (Stanley et al., 2018). Figure 2 shows the location of these four measurement stations: Ridge Hill in the West of England, Angus on the East coast of Scotland, Tacolneston on the East coast of England and Mace Head on the West coast of Ireland. Measurements are made quasi-continuously throughout this network, but are averaged into hourly values here, consistent with the time step of the meteorology that drives the atmospheric transport model (section 3.1.2). The data set contains $\sim$10 000 measurements to demonstrate the capabilities of the method at handling moderate data volumes. We consider the scalability of this method in the discussion (section 4).

#### 3.1.2 Transport model

An atmospheric transport model calculates the sensitivity of hourly measurements to the emissions or boundary conditions, from which the matrices $\mathbf{H}$ and $\mathbf{K}$ can be formed. This work uses the NAME III (Numerical Atmospheric dispersion Modelling Environment) version 7.2 Lagrangian particle dispersion model (Jones et al., 2006) to simulate the transport of methane in the atmosphere. For each measurement, NAME tracks 20 000 gas particles, released over a one hour period, backward in time for

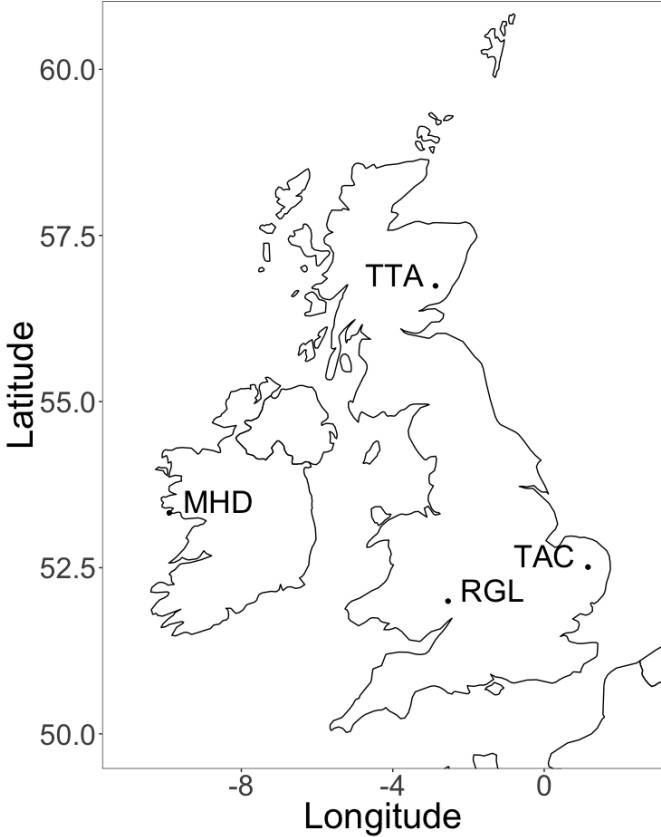

**Figure 2.** The case studies use the four UK DECC network measurement stations located on this map. RGL is Ridge Hill; TAC is Tacolneston, TTA is Angus and MHD is Mace Head station.

days from the measurement site. We record the times and locations that the particles drop below 40 magl and reach the computational domain boundary, which is at 5° S, 74° N, 55° E and 192° E, to calculate the sensitivity of the data to emissions from the surface or to the mole fraction at the domain edge. NAME was driven by the Met Office's Unified Model UK Variable (UKV) three hourly meteorological analysis (Cullen, 1993).

### 3.1.3 Prior emissions inventory

Inventory data from the Emissions Database for Global Atmospheric Research (EDGAR) v4.3.2. provides the mean prior emissions (Janssens-Maenhout et al., 2017) using the most recent emissions map from 2012. This database contains anthropogenic emissions, which are the dominant methane sources in the UK (Manning et al., 2011), and is deemed appropriate for the application. The EDGAR emissions inventory provides an a priori estimate of UK methane emissions of 2.46 Tg/yr. The MOZART-4 global chemistry model (Emmons et al., 2010), driven by global fields of natural and anthropogenic emissions and

sink terms (Ganesan et al., 2017), provides the prior mean estimate of methane mole fraction at the boundaries of the inversion region.

## 3.2 Pseudo-data experiment

We test the method by performing an inversion using pseudo-data for four consecutive time periods of one month. By creating a known emissions field we are able to validate the method through comparing the inferred emissions to the known emissions, which is not possible in the real world. We form a synthetic emissions field by allowing the emissions to deviate from the prior mean emission according to a Matérn field (see section 2.2). We choose to simulate the data using $\sigma = 0.5$, as the uncertainty in the EDGAR v4.3.2. inventory is around 50% for methane emissions, and $\rho = 3.25$, which is similar to the correlation length scale in UK emissions in the EDGAR v4.3.2. inventory estimated using a variogram (Cressie, 2015), although the correlation length scale in the uncertainty is unknown. We use $\phi = 0.8$ as expert experience suggests that UK emissions of methane are generally highly correlated in time.

To create the synthetic emissions field, the NAME sensitivities to measurements to the measurement stations detailed in section 3.1.1 at each grid cell is multiplied by the corresponding EDGAR inventory emissions in that grid cell, which are then transformed into the triangulation nodes in Figure 1. This forms the matrix $\mathcal{H}$. We randomly sample the full spatio-temporal precision matrix for the latent field (section 2.3) using the GMRFLib library (Rue and Follestad, 2001) to generate the latent field $\mathbf{x}$. In this experiment we treat the boundary conditions as known as in practice these are generally well constrained (or treated as known) during an inversion. The observations are simulated using the simulated latent field and sensitivities following section 2.1, with additive Gaussian noise with a standard deviation of 15 % of $\mathbf{y}$. The synthetic observations contain a total of 11 520 measurement points, which are used to infer 646 emissions nodes for each time period, i.e. the nodes of the mesh in Figure 1. Figures 3a-3d show the synthetic emissions field in terms of deviation from some prior emissions field. The synthetic total deviation in emissions from the prior mean for the UK for each period is 0.07, 1.29, 1.01 and -0.33 Tg/yr.

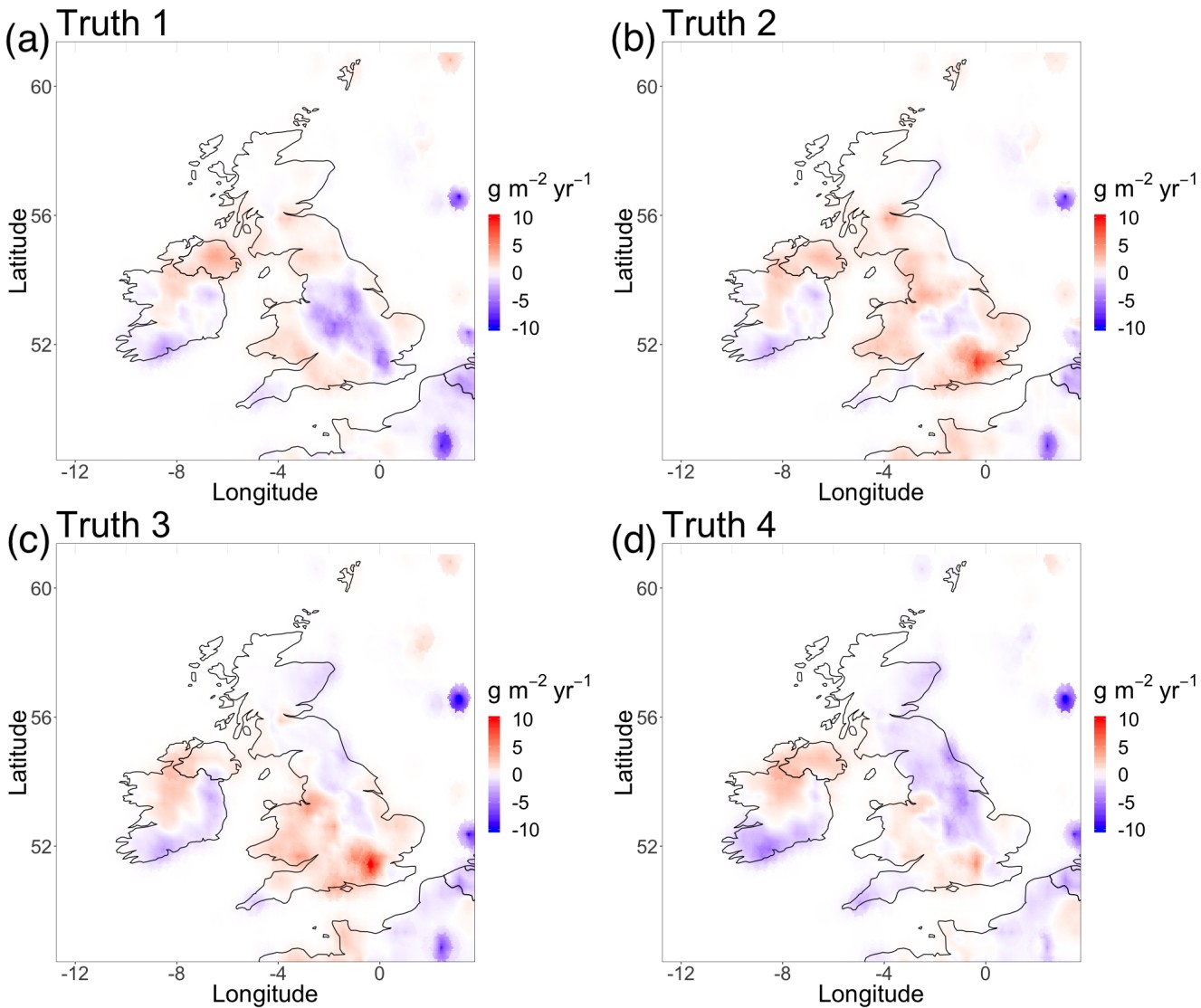

**Figure 3.** Time correlated deviation of emissions from the prior mean, simulated using a Matérn field for (a) the first time period, (b) the second time period, (c) the third time period and (d) the fourth time period.

The inference needs prior probabilities for the hyperparameters, which are known exactly here, but we set them to be deliberately incorrect, but feasible based on true prior knowledge, to check that the inversion method can still recover the correct emissions. For the inversion we assign a prior probability for $\phi$ using $a$=6.5 and $b$=0.1, for $\sigma$ using $\sigma_0 = 0.1$ and $p_\sigma = 0.01$, and $\rho$ using $\rho_0 = 5$ and $p_\rho = 0.5$. We base the constraint on the spatiotemporal emissions on the assumption that methane emissions in the UK are likely to be strongly correlated between time periods, are unlikely to vary by more than 10 % of the a priori emissions from the previous time step and that there is little knowledge of the spatial correlation structure.

For the model measurement error we assign the prior probability on the log precision as $\log \frac{1}{\sigma_y^2} \sim \mathcal{N}(-5, 1)$, which represents approximate 68 % probability that the error falls between 8 and 20 ppb, in line with previous experience.

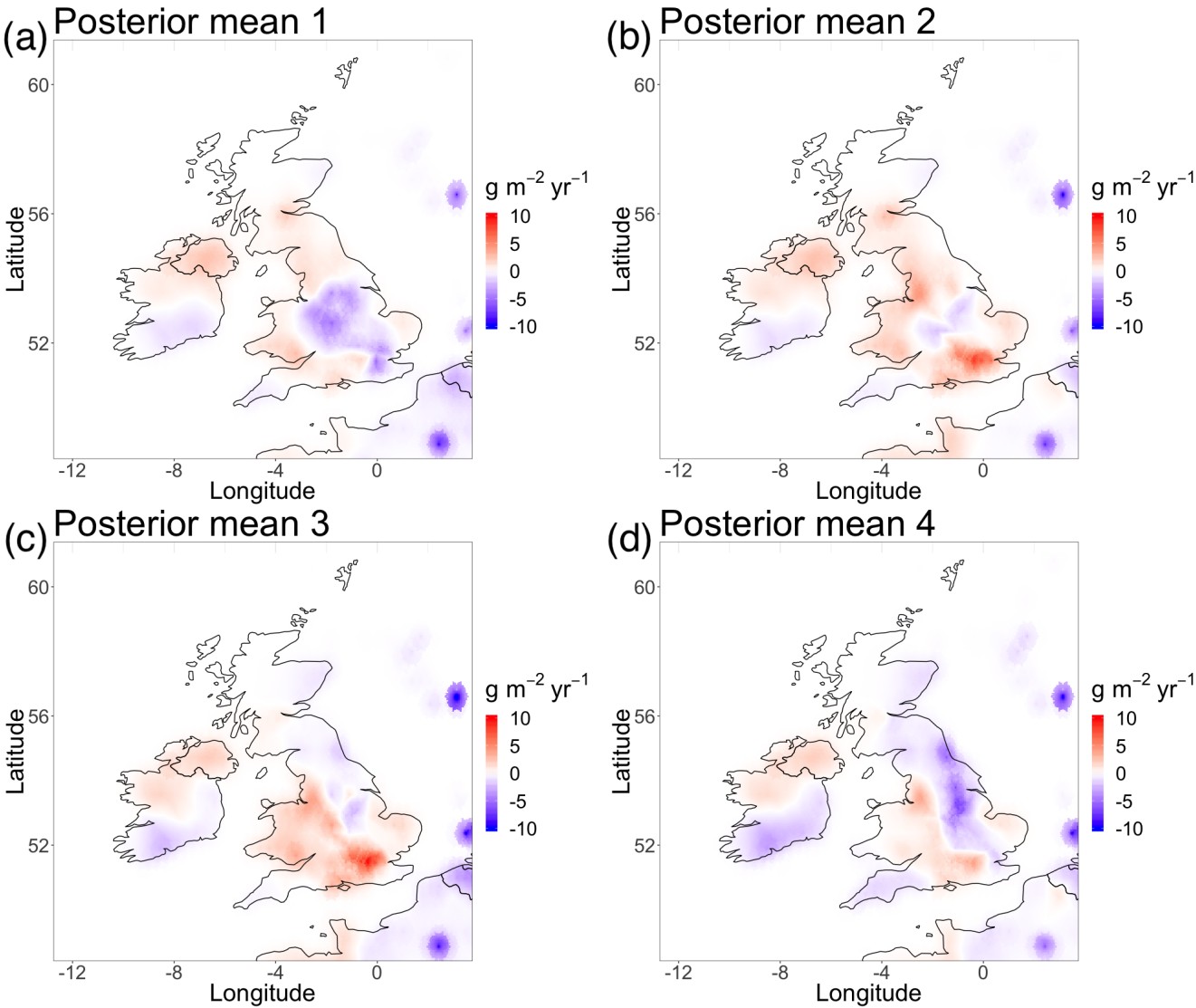

**Figure 4.** The inferred mean deviation of emissions from the prior mean for (a) the first time period, (b) the second time period, (c) the third time period and (d) the fourth time period.

Figure 4 shows the resultant inferred mean change in the emissions field for each time period using INLA (section 2.5) and the hierarchical model in section 2.4. The change in the emissions field is the inferred latent field scaling parameter multiplied by the inventory emissions in section 3.1.3. Figure S1 shows maps of the 95% uncertainty in the difference in emissions. A linear combination of the posterior latent field multiplied by the inventory emissions field gives total emissions for the UK.

The inferred posterior mean change in total emissions for the UK from the prior mean with their associated 95 % uncertainties for the four time periods are 0.01 [-0.10, 0.12], 1.23 [1.10, 1.36], 1.07 [0.97, 1.18] and -0.30 [-0.41, -0.18] Tg/yr, which are shown in Figure 5. The inferred emissions for the UK as a whole agree well with the synthetic emissions, with all true synthetic emissions totals falling within the 95 % uncertainty of the inferred total emissions. These synthetic tests cover both small and

5    large differences between the a priori estimate of methane emissions in the UK and its true value. The larger differences are most likely an overestimate of what would be expected in practice, although this provides a good test case. The posterior mean spatial distribution of emissions in figure 4 are qualitatively similar to those in figure 3, although we avoid reading heavily into comparisons of the mean estimates plotted on spatial maps (see Gelman and Price, 1999).

Hyperparameter estimation is less accurate than for the latent field. The estimation of the noise $\sigma_y$ generally captures the

10    imposed noise well, which had a mean value of 5.5 ppm, with an estimated value of 6.6 [6.5, 6.8] ppm. The correlation structures are less well captured. The temporal correlation $\phi$ was estimated with a mean value 0.7 [0.6, 0.8], the range $\rho$ has value 1.7 [1.3, 2.1] and the marginal standard deviation of the latent field $\sigma$ with the value 0.8 [0.7,0.9]. It is promising that the posterior mean estimates of all hyperparameters show an improvement on their prior mean or baseline values, although only the true value for $\rho$ falls within the estimated 95% uncertainty.

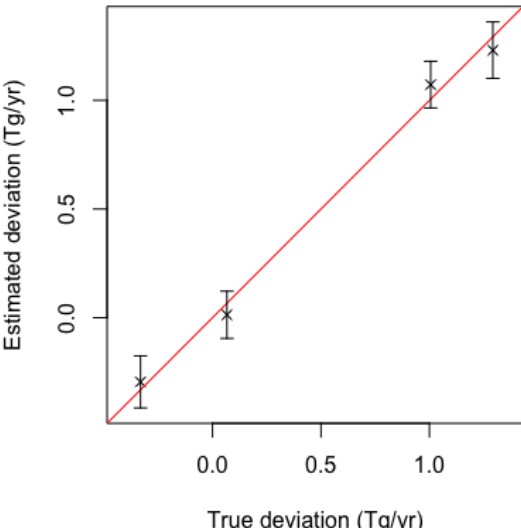

**Figure 5.** The inferred and true difference between UK emissions and a synthetic inventory value for each of the four time periods. The crosses show the inferred mean difference in total emissions with their associated 95 % uncertainty. The red line indicates one to one agreement.

### 3.3 Real data experiment

This section presents 2014 methane emissions estimates for the UK. The year is split into four time periods: January to March, April to June, July to September and October to December. We use the same prior probabilities for the hyperparameters as the inversion using the synthetic dataset in section 3.2. The prior boundary conditions are distributed with $\mu_{\sigma_{BC}} = 3.2$ and $\sigma_{\sigma_{BC}} = 0.4$, based on expert judgment. The a priori value for these boundary conditions come from the MOZART-4 model as 3.1.3. The linear mapping from the latent field to the measurements is generated using the NAME-derived sensitivities described in section 3.1.2 multiplied by the the inventory emissions detailed in section 3.1.3.

Figure 6 shows maps of the inferred mean difference in emissions from the a priori inventory for the four time periods using INLA as section 2.5. This result is the mean posterior scaling for the latent field multiplied by the inventory value (see section 3.1.3). Figure S2 shows maps of the 95% uncertainty in the difference in emissions. A linear combination of the posterior latent field multiplied by the inventory emissions field gives total emissions for the UK. We estimate UK-total methane emissions in 2014, with their associated 95% uncertainty, of 2.14 [1.90, 2.38], 2.28 [2.06, 2.51], 2.62 [2.39, 2.84] and 2.09 [1.87, 2.30] Tg/yr for the periods January to March, April to June, July to September and October to December, respectively. These are plotted along with the inventory estimate in Figure 7. This emissions trend suggest that for 2014 there was an increase in methane emissions in the UK during the summer months compared to the winter months. The uncertainties, however, are large, meaning that this increase may not be as stark as suggested by the mean estimates. Combining these emissions into a mean annual emission for 2014 with its associated two standard deviation uncertainty, assuming that time periods are correlated with the modal posterior value of $\phi$, gives $2.28 \pm 0.33$ Tg/yr. The emissions are similar to mean UK estimates from previous hierarchical inversions using NAME of 2.09 [1.65, 2.67] Tg/yr by Ganesan et al. (2015) and 2.28 [2.04, 2.52] Tg/yr by Lunt et al. (2016). All of these estimates are broadly in line with the United Kingdom's 2014 emissions estimate of 2.13 Tg/yr reported in its national inventory (Department of Business, Energy and Industrial Strategy, 2019).

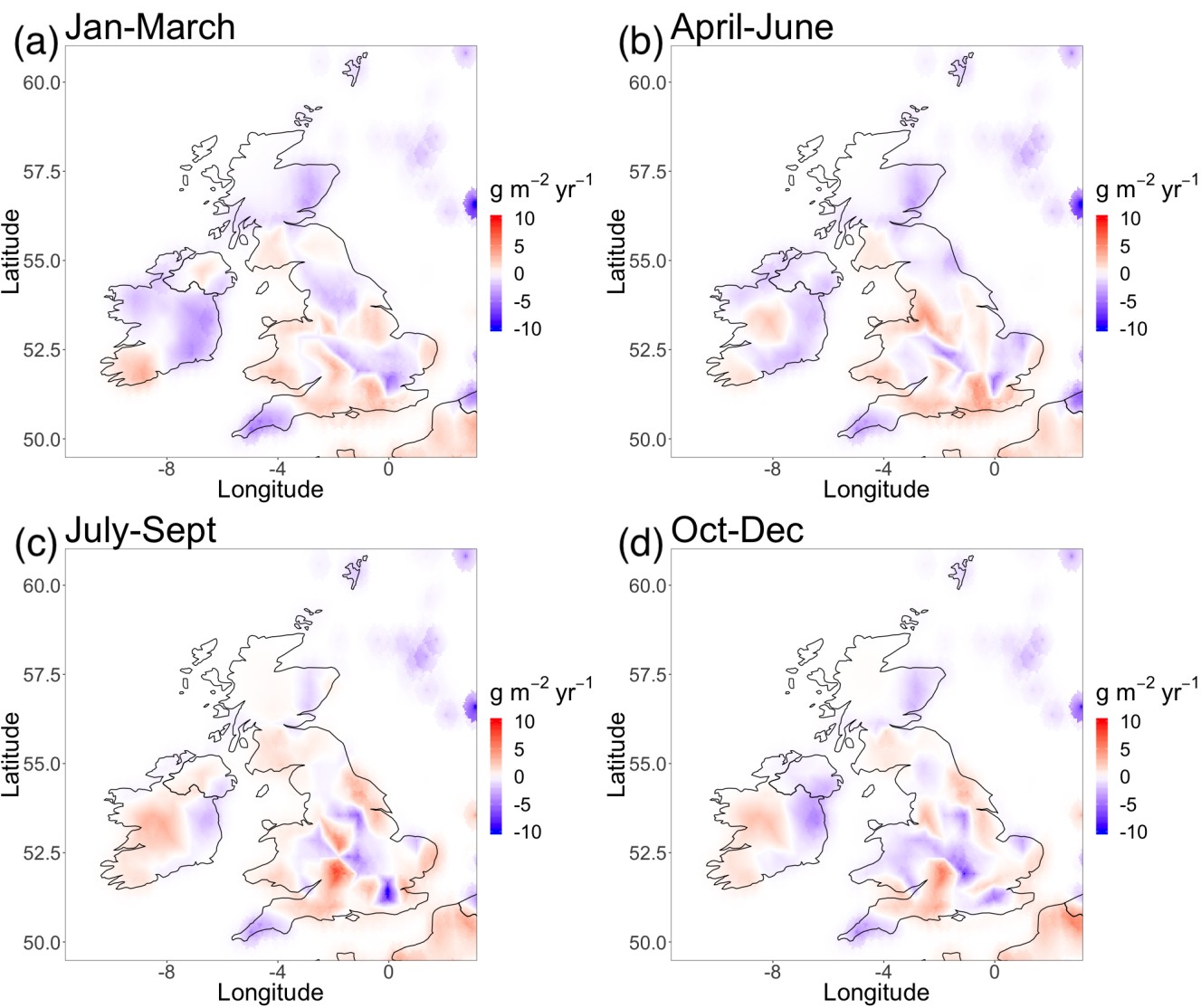

**Figure 6.** Inferred mean difference in methane emissions for the UK in 2014 compared to the EDGAR inventory for (a) January to March, (b) April to June, (c) July to September and (d) October to December.

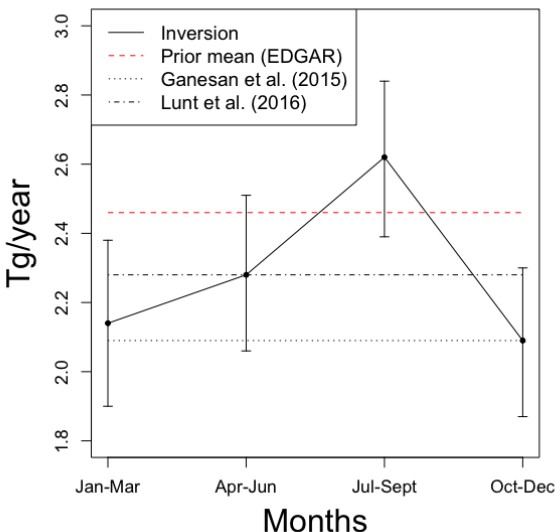

**Figure 7.** Estimated total UK methane emissions for 2014. The prior mean comes from EDGAR v 4.32 inventory data (red dashed). For comparison, the figure also shows the posterior means for UK methane emissions from Ganesan et al. (2015) (black dotted) and Lunt et al. (2016) (black dash-dotted).

## 4    Discussion

The real benefit of the presented inversion method is speed, while still maintaining the idea that, in this application, uncertainties exist within the uncertainties that are inherent to hierarchical Bayesian inverse methods. The computation of the marginal posterior distribution of the latent field is readily suited to run in parallel across multiple cores, making this approach scalable

5    to problems with a larger parameter space. This, however, requires that sufficient memory allocation is available. The inference for the experiment in section 3.3 took around two hours wall-clock time using a four core MacBook Pro with 8GB RAM. For comparison, we were unable to get the problem in section 3.3 to converge using 22 million iterations of a Metropolis-adjusted Langevin diffusion MCMC algorithm (Roberts and Rosenthal, 1998), taking on the order of days to complete and whose convergence is difficult to diagnose (Cowles and Carlin, 1996). Reducing the problem to purely spatial inference reduces the

10    run time down to around 10 minutes wall-clock time for a single three month period. Inference using INLA may have smaller computational gains over hierarchical methods other than MCMC, for example empirical hierarchical inference (e.g. Michalak et al., 2005). Such methods, however, would benefit from using the stochastic partial differential equation approach to GMRFs for spatial correlation in 2.2, vastly reducing the cost of inverting dense covariance matrices.

    The latent Gaussian field, crucial to this method, has the problem that it does not restrict emissions to strictly positive

15    values, which is a physical requirement for many gas emissions. In addition, the INLA method relies on the assumption of

approximate multivariate normality of the posterior linear predictors. MCMC algorithms do not suffer from this issue, as any prior probability density function can be chosen and no assumption is made about the posterior linear predictors. This has to be traded off, however, with the speed and ease of implementation of the method and the scale of the problem that the user wishes to solve. If strictly positive posterior emissions is an absolute requirement, another possible modification to the approach is to use a Taylor expansion around the nonlinear model $\mathbf{z} = \mathbf{H}e^{\mathbf{x}} + \epsilon$, where $\mathbf{x}$ is a multiplicative scaling of the prior mode which follows a Matérn field. This Taylor expansion could either be around zero if the prior inventory is a good estimate of the true emissions, or around the posterior mode itself, which can be found through iterations of the inverse method at the posterior mode of the previous iteration. An alternative route would be to instead use a non-Gaussian Matérn field and this will likely prove a promising future avenue for geostatistical inference (Wallin and Bolin, 2015).

A potential extension to this work is global scale modelling, for example a global study of methane emission from global satellite measurements. Spatiotemporal estimation of global $CO_2$-fluxes using GMRFs and a global measurement network gives promising results (Dahlén et al., 2019). The nature of the method makes it well-suited to parallel implementation and thus up-scaling. Up-scaling to global studies may introduce additional difficulties, which have been otherwise ignored in the regional case. For example it may no longer be possible to assume a spatial correlation structure controlled by only two hyperparameters, i.e. the correlation structure may vary in space.The stochastic partial differential equation approach to Gaussian Markov random fields can handle this by allowing the hyperparameters $\rho$ and $\sigma$ to become vectors, or a non-stationary covariance (Lindgren et al., 2011), or by subsetting the space controlled by the different covariance structures (Sha et al., 2019). The problem of non-stationary covariances may also be present for inference of on other spatial scales. For example natural features, such as lakes, may cause abrupt changes in correlation structures. Emissions of anthropogenic greenhouse gases may exhibit no spatial correlation structure (e.g. Mühle et al., 2019; Rigby et al., 2019). In this case a Matérn field would be inappropriate, but an autoregressive model may still be applicable for time-varying emissions. The inclusion of a non-stationary covariance makes the problem much more computationally expensive and a more parsimonious approach may often suffice (Fuglstad et al., 2015).

## 5 Conclusions

This work presents a fast and efficient method using an integrated nested Laplacian approximation for hierarchical inference of trace gas emissions. This method is particularly well-suited to assimilating large data sets. We show that INLA with a stochastic partial differential equation approach for spatial correlation can reproduce synthetic emissions from pseudo-observations and bench-marked emissions using real data.

A real advantage over other hierarchical Bayesian inversion methods is the attractive convergence properties, which can be difficult to obtain using methods such as Markov chain Monte Carlo algorithms. As the method computes the marginal variance for each node, this allows for efficient parallel implementation and significant computational savings compared to other hierarchical methods. The importance of computational speed will become increasingly important as more data from space-borne sensors become available, which will offer an increased number of measurements and spatial coverage.

*Code and data availability.* Measurements of methane from the UK DECC network sites Tacolneston, Ridge Hill and Tall Tower Angus are available at ebas.nilu.no. Measurements of methane for the Mace Head station are available at http://agage.eas.gatech.edu/data_archive/. The NAME III v7.2 transport model is available from the UK Met Office under licence by contacting enquiries@metoffice.gov.uk. The meteorological data used in this work from the UK Met Office operational NWP (Numerical Weather Prediction) Unified Model (UM) is available from the UK Centre for Environmental Data Analysis at http://data.ceda.ac.uk/badc/ukmo-nwp. The MOZART-4 global chemistry model is available at https://www2.acom.ucar.edu/gcm/mozart-4. The R-INLA v17.06.20 library is available for download from http://www.r-inla.org/download, which includes the GMRFLib library. The EDGAR v.4.3.2 methane inventory can be downloaded from https://edgar.jrc.ec.europa.eu/overview.php?v=432_GHG. The code and data to infer emissions using the simulated data in section 3.2 can be found at https://osf.io/53w96, DOI 10.17605/OSF.IO/53W96.

Any further data or code is available from the corresponding author on request.

*Author contributions.* LW and ZS conceived and lead the implementation of the study. MR, AG and JR supported and advised on the study. KS, SO and DY made the measurements from the UK DECC network. AM created the sensitivity footprints using the NAME model. LW lead the writing of the manuscript, to which all authors have contributed and edited.

*Competing interests.* There are no competing interests.

*Acknowledgements.* This work was funded by the Jean Golding Institute Seed Corn grant CHEM.HF8064. Luke Western was funded by grants NE/M014851/1 and NE/S016155/1 from the Natural Environment Research council, and the a grant from the UK Department for Business, Energy and Industrial Strategy. Anita Ganesan was funded under a Natural Environment Research Council Independent Research Fellowship NE/L010992/1.

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
