# Peer review of "Bayesian spatiotemporal inference of trace gas emissions using an integrated nested Laplacian approximation and Gaussian Markov random fields"

_Geoscientific Model Development, 2019_

## Referee Comment (RC1) · Alfredo Farjat (Referee) · 23 Aug 2019

General comments The authors present a model for inferring spatially and temporal correlated emissions of greenhouse gases. The model is constructed following a Bayesian hierarchical framework and the Integrated Nested Laplace Approximation (INLA) method is used for making inferences. The stochastic partial differential equation (SPDE) approach is exploited with a Gaussian Markov Random Field (GMRF) representation. The main advantage of the approach presented is the computational efficiency for Bayesian inference. So it represents an alternative to MCMC methods
that tend to be slow and hard to achieve convergence in the presence of multiple parameters. The concept and ideas presented represent and contribution. However, the presentation of methods and results can be improved and sharing the codes developed would help others to use the proposed method. Please see below my specific comments and suggestions to improve the manuscript.

Specific comments Page 3: In Section 2.1 the linear model for the spatial field is briefly described and derived. This section can be improved without loss of generality by starting from equation 3 that contains all relevant quantities of interest expressed as deviations from prior mean values. In addition, the physics behind factor x or some interpretation about its connection with the measurements could be added in this section to help the reader better understand the model. Page 4, Line 2: The solution to the stochastic differential is a stationary Gaussian Field with Matern covariance structure. The fact that the process is stationary should be mentioned. Moreover, if the covariance function does not depend on the direction but just on the Euclidean distances between $s\_i$ and $s\_j$, then it should also be mentioned that the process is isotropic. Page 5, equation 6: The indices and ranges should be explained. For instance, i=1,..,n. Does n represent the number of nodes in the mesh? Also, index j goes from 1 to 4. Does this represent the sides of the rectangular region? Page 7, Line 15: Is there any advantage regarding the computational cost or convergence for using penalized complexity priors instead of a vague prior? Pages 10-13, Section 3.2: More information about the simulation study should be included in the main text or supplementary materials. Would be nice to see the relationship between the true parameters from the simulated data and the estimated values along with their corresponding uncertainty. Also, would be interesting to complement the posterior mean estimate plots as function of time with analogous plots showing some measure of the uncertainty on the estimates (e.g. SE or credible set). Last, the value of this manuscript would be greatly improved if the R code used for the numerical experiment could be shared. However, if the complexity and scale of the problem makes this option not feasible, then at least a toy example to illustrate the concept would be useful. Pages 15-16: The discussion section could

mention some limitations of the method. First, the stationarity assumption may not be suitable when modelling some environmental phenomena. Particularly in atmospheric phenomena it may inappropriate to assume that the spatial correlation is the same throughout the domain as topographical variables (e.g. mountains, lakes, etc.) might have an influence on the spatial dependence. Second, INLA method relies on the assumption of approximate multivariate normality of the posterior linear predictors.

Technical corrections Page 4, Line 19: provided instead of providing. Page 6, Line 12: consider rephrasing: "... which means that the spatiotemporal precision matrix can be expressed using a Kronecker product as follows...", and delete Line 15. Page 6, Line 17: burden instead of size. Page 6, Line 19: consider rephrasing: "...constant over a three-month period..." Page 6, Line 19-20: Later in the text you use four three-month periods, not three. Page 7, Line 10: consider rephrasing "...for the spatial precision matrix Q(\theta)ˆ{-1}, are ..." Page 16, Line 13: rephrase to "...correlation structure may vary in space."
* * *

---

## Referee Comment (RC2) · Anonymous Referee #2 · 20 Sep 2019

Main comments: This article describes and evaluates a new approach to hierarchical Bayesian inversion that can improve the computational efficiency of the posterior sampling of trace gas emissions and hyperparameters used to define their prior probability distribution (pdf). While the proposed method is appealing to address current challenges in regional greenhouse gas inversions when a general framework is adopted (e.g., non-gaussian pdf, unregular mesh), the paper lacks a sufficient level of details in the description of the technique. Those methodological details are of significant importance for a journal such as GMD, which focuses on technical aspects of model

developments. Also, since the integrated nested Laplacian approximation seems to be the core of the methodological contribution of this paper, it is difficult to understand why its description is missing (even if proper references are given). Provided such information is added to the manuscript (see specific comments below), this article should be suitable for publication in GMD.

Specific comments: - P3, L9: "...which maps the surface emissions (x) to the measurements...". The variable x was not defined here. - P3, L16: I do not understand the discussion about the change of variable in x (which is simply a shift in the distribution), and in particular how this relates to the positiveness of the fluxes. It seems one issue here could be that a log-normal distribution may be more suitable than a Gaussian one (if positiveness needs to be enforced). Apart from a re-centering of the prior pdf around zero, I do not see what the transformation achieves. Please clarify. - P6, L1: "...measurements made at a given time are independent...". Would it be easy to generalize the approach for spatially correlated observations at a given time? This, for instance, would be useful for inversions based on satellite measurements. - P6, L6-9: Di you mean "emissions" instead of "measurements" here? You mention measurements (y) but then refer to emissions (x) in the equation. Please clarify. Also, could you briefly comment on the form taken by the error covariance in (8b)? For instance, the presence of phi here and its role is not very intuitive. - Section 2.5: This section should be entirely rewritten and much more details have to be provided. For instance, please explain the posterior sampling approach adopted here as well as the principle of the integrated nested Laplacian approximation used. This seems to be the core of the methodological contribution of this paper.

---

## Author Comment (AC1) · 14 Oct 2019

We would like to thank the reviewers for their helpful and supportive comments on our discussion article 'Bayesian spatiotemporal inference of trace gas emissions using an integrated nested Laplacian approximation and Gaussian Markov random fields' (gmd-2019-66). The revised manuscript, along with new supplementary figures, are attached with revised sections indicated in red. Below we address each reviewer's comments in turn, where the comment has been italicised.

[Figure]

**Reviewer 1**

*Page 3: In Section 2.1 the linear model for the spatial field is briefly described and derived. This section can be improved without loss of generality by starting from equation 3 that contains all relevant quantities of interest expressed as deviations from prior mean values. In addition, the physics behind factor x or some interpretation about its connection with the measurements could be added in this section to help the reader better understand the model.*

We have addressed this comment by reworking most of section 2.1 (P3 L4-23). A physical interpretation of $x$ has been added and the linear model starts from the previous equation 3 (now equation 1), which removes the need for the tilde notation in the remainder of the manuscript. The main additions are:

(L6-8) 'The a priori emissions at given location are generally informed by spatially resolved bottom-up inventories (as in section 3.1.3) or extrapolation from some reported emissions value. The value $\mathbf{x}$ is a multiplicative scaling of this a priori value for emissions, most generally expressed as a quantity of gas per unit time per unit area.'

(L13-23) 'This is in an effort to fit the physical model to the imposed statistical model that fast computation requires. Taking the approach that, for many regional inverse problems involving longer-lived trace gases, there is a linear relationship between emissions which are constant in time and observed atmospheric concentration, the relationship between measurements and emissions is,

$$\mathbf{y} = \mathbf{H}\mathbf{x} + \mathbf{K}\mathbf{u} + \epsilon, \tag{1}$$

where $\mathbf{y}$ is a vector of the residual between the measured and a priori predicted measurement, $\mathbf{H}$ is a Jacobian (or sensitivity) matrix, which maps the surface emissions to the measurements, $\mathbf{u}$ is a vector of independent and identically distributed variables containing the contribution to the measurement of mole fractions

at the boundary of the domain minus the prior mean contribution, with an associated sensitivity matrix $\mathbf{K}$, and $\epsilon$ is some stochastic error. The variable $\mathbf{x}$ has a Gaussian prior probability with zero expectation and a covariance described by a Gaussian random field. This will be solved using a hierarchical framework (section2.4).'

*Page 4, Line 2: The solution to the stochastic differential is a stationary Gaussian Field with Matern covariance structure. The fact that the process is stationary should be mentioned. Moreover, if the covariance function does not depend on the direction but just on the Euclidean distances between s_i and s_j, then it should also be mentioned that the process is isotropic.*

The stationarity has been mentioned (P3 L28) as well as elaboration of the process being isotropic, with further reference to information on non-stationary and anisotropic process (P4 L3-4) '...noting that that the dependence is only on the Euclidean distance and so the process is isotropic. For an example of non-stationary and anisotropic fields see e.g. Marques et al. (2019).'

*Page 5, equation 6: The indices and ranges should be explained. For instance, i=1,..,n. Does n represent the number of nodes in the mesh? Also, index j goes from 1 to 4. Does this represent the sides of the rectangular region?*

This notation has now been clarified (P5 L6) '$i$ represents each of the total $n$ nodes and $j$ represents the edge of the domain at each of the four cardinal directions'.

*Page 7, Line 15: Is there any advantage regarding the computational cost or convergence for using penalized complexity priors instead of a vague prior?*

Penalised priors do not increase the computational speed for the given case and are chosen instead for their intuitiveness. This has been added to the manuscript (P7 L 19).

*Pages 10-13, Section 3.2: More information about the simulation study should be included in the main text or supplementary materials. Would be nice to see the relationship between the true parameters from the simulated data and the estimated values along with their corresponding uncertainty.*

We have added a discussion on the estimated hyperparameters for the simulated data (P14 L4-9), 'Hyperparameter estimation is less accurate than for the latent field. The estimation of the noise $\sigma_y$ generally captures the imposed noise well, which had a mean value of 5.5 ppm, with an estimated value of 6.6 [6.5, 6.8] ppm. The correlation structures are less well captured. The temporal correlation $\phi$ was estimated with a mean value 0.7 [0.6, 0.8], the range $\rho$ has value 1.7 [1.3, 2.1] and the marginal standard deviation of the latent field $\sigma$ with the value 0.8 [0.7,0.9]. It is promising that the posterior mean estimates of all hyperparameters show an improvement on their prior mean or baseline values, although only the true value for $\rho$ falls within the estimated $95\%$ uncertainty.'

*Also, would be interesting to complement the posterior mean estimate plots as function of time with analogous plots showing some measure of the uncertainty on the estimates (e.g. SE or credible set).*

These plots, showing the 95% uncertainty, have been added as supplementary plots (P25-26).

*Last, the value of this manuscript would be greatly improved if the R code used for the*

*numerical experiment could be shared. However, if the complexity and scale of the problem makes this option not feasible, then at least a toy example to illustrate the concept would be useful.*

We have made the code and data for the simulated data example freely available via the Open Science Foundation (osf.io/53w96, DOI 10.17605/OSF.IO/53W96), which has been noted in the Code and data availability section of the manuscript.

*Pages 15-16: The discussion section could mention some limitations of the method. First, the stationarity assumption may not be suitable when modelling some environmental phenomena. Particularly in atmospheric phenomena it may inappropriate to assume that the spatial correlation is the same throughout the domain as topographical variables (e.g. mountains, lakes, etc.) might have an influence on the spatial dependence. Second, INLA method relies on the assumption of approximate multivariate normality of the posterior linear predictors.*

We have added additional discussion on these topics in section 4, in particular P18, L16-20. 'The problem of non-stationary covariances may also be present for inference of on other spatial scales. For example natural features, such as lakes, may cause abrupt changes in correlation structures. Emissions of anthropogenic greenhouse gases may exhibit no spatial correlation structure (e.g. Mühle et al., 2019; Rigby et al., 2019). In this case a Matérn field would be inappropriate, but an autoregressive model may still be applicable for time-varying emissions.'

*Page 4, Line 19: provided instead of providing.*
This has been changed.
*Page 6, Line 12: consider rephrasing: '. . . which means that the spatiotemporal*

*precision matrix can be expressed using a Kronecker product as follows...', and delete Line 15.*
This has been changed.
*Page 6, Line 17: burden instead of size.*
This has been changed.
*Page 6, Line 19: consider rephrasing: '... constant over a three-month period ...'*
This has been changed.
*Page 6, Line 19-20: Later in the text you use four three-month periods, not three.*
This has been changed to three-month
*Page 7, Line 10: consider rephrasing '... or the spatial precision matrix $Q(\theta)^{-1}$, are ...'*
This has been changed.
*Page 16, Line 13: rephrase to '... correlation structure may vary in space.*
This has been changed.

**Reviewer 2**

*P3, L9: '... which maps the surface emissions (x) to the measurements ...'. The variable x was not defined here.*

This is now defined in section 2.1, L7-8. 'The value $x$ is a multiplicative scaling of this a priori value for emissions, most generally expressed as a quantity of gas per unit time per unit area.'

*P3, L16: I do not understand the discussion about the change of variable in x (which is simply a shift in the distribution), and in particular how this relates to the positiveness of the fluxes. It seems one issue here could be that a log-normal distribution may*

*be more suitable than a Gaussian one (if positiveness needs to be enforced). Apart from a re-centering of the prior pdf around zero, I do not see what the transformation achieves. Please clarify.*

The use of GMRFs are a requirement for the computational efficiency of the method. Therefore, we have attempted to fit a statistic model to the physical model. We have clarified this point in the reworked section 2.1, in particular P3 L13. 'This is in an effort to fit the physical model to the imposed statistical model that fast computation requires.'

*P6, L1: '... measurements made at a given time are independent ...'. Would it be easy to generalize the approach for spatially correlated observations at a given time? This, for instance, would be useful for inversions based on satellite measurements.*

This could be generalised but this correlation would have to result in a sparse covariance matrix (e.g. AR1, Matern GMRF) to keep the computational gains. We have noted this on P6 L1-2. 'This can be generalised to include correlated measurements, where the covariance function should result in a sparse matrix to retain computational efficiency. '

*P6, L6-9: Do you mean 'emissions' instead of 'measurements' here? You mention measurements (y) but then refer to emissions (x) in the equation. Please clarify.*

Thank you for noticing this. The sentence has now been reworded to 'The time varying structure of $\mathbf{H}_t$ and $\mathbf{x}$ applies also to $\mathbf{K}_t$, and $\mathbf{u}_t$.' We have elaborated on the use of an autoregressive model and the parameter $\phi$ on P6 L11-14. 'We have chosen an autoregressive model of order one as we believe that emissions are generally similar to those at previous time step. In the given model, $\phi$ is the correlation between the

previous time step and the current time step. This means that the emissions at time $t$ have a similarity of $\phi$ to emissions at $t-1$ plus some spatially correlated random effect $\delta_t$.'

*Section 2.5: This section should be entirely rewritten and much more details have to be provided. For instance, please explain the posterior sampling approach adopted here as well as the principle of the integrated nested Laplacian approximation used. This seems to be the core of the methodological contribution of this paper.*

The section 2.5, P6, has been entirely rewritten with much more detail about the posterior sampling approach, outlining the principles of INLA.

An integrated nested Laplacian approximation (Rue et al., 2009) provides a fast and efficient framework to infer the latent variable $\mathbf{x}$ and hyperparameters $\boldsymbol{\theta}$ from measurements $\mathbf{y}$. The calculation of the INLA is possible using the R-INLA package (Lindgren & Rue, 2015). In this work we use R-INLA version 17.06.20. The speed in this approach comes from solving the marginal posteriors for $x_i$ through the numerical integration

$$(\theta_j \mid \mathbf{y}) = \int (\boldsymbol{\theta} \mid \mathbf{y}) \mathrm{d}\boldsymbol{\theta}_{-j}, \tag{2}$$

$$(x_i \mid \mathbf{y}) = \int (x_i \mid \mathbf{y}, \boldsymbol{\theta})(\boldsymbol{\theta} \mid \mathbf{y}) \mathrm{d}\boldsymbol{\theta}. \tag{3}$$

Equations 2 and 2 make use of a Laplace approximation, by approximating $(\boldsymbol{\theta} \mid \mathbf{y})$ by

$$(\boldsymbol{\theta} \mid \mathbf{y}) \propto \left. \frac{(\mathbf{x}, \boldsymbol{\theta}, \mathbf{y})}{_G(\mathbf{x} \mid \boldsymbol{\theta}, \mathbf{y})} \right|_{\mathbf{x}=\mathbf{x}^*(\boldsymbol{\theta})}, \tag{4}$$

where $_G(\mathbf{x} \mid \boldsymbol{\theta}, \mathbf{y})$ is a Gaussian approximation to the full conditional of $\mathbf{x}$ and $\mathbf{x}^*(\boldsymbol{\theta})$ is the modal probability of $\mathbf{x}$ for a given $\boldsymbol{\theta}$. This approximation is exact if $(\boldsymbol{\theta} \mid \mathbf{y})$ is

Gaussian, and gives a good approximation for log concave problems (Tierney and Kadane, 1986). Then, it is possible to approximate $(x_i \mid \mathbf{y}, \boldsymbol{\theta})$ using another Laplace approximation

$$_{LA}(x_i \mid \boldsymbol{\theta}, \mathbf{y}) \propto \left.\frac{(\mathbf{x}, \boldsymbol{\theta}, \mathbf{y})}{_G(\mathbf{x}_{-i} \mid x_i, \boldsymbol{\theta}, \mathbf{y})}\right|_{\mathbf{x}_{-i} = \mathbf{x}^*_{-i}(x_i, \boldsymbol{\theta})}, \tag{5}$$

where $_G(\mathbf{x}_{-i} \mid x_i, \boldsymbol{\theta}, \mathbf{y})$ is the Gaussian approximation to $\mathbf{x}_{-i} \mid x_i, \boldsymbol{\theta}, \mathbf{y}$ evaluated at the mode $\mathbf{x}^*_{-i}(x_i, \boldsymbol{\theta})$. See Rue et al. (2009) and Martins et al. (2013) for a more in depth description.

While this method relies on calculating only the marginal posterior distributions of $x_i$, it is still possible to predict a linear combination of the field to provide regional emissions totals (e.g. country totals). We define a linear predictor of emissions for a given region $\eta*$, defined using the basis function representation of the mesh where each node contains information on the spatial area represented by that node and its connecting vertices and zeros for all nodes that are not in the region of interest. Then we can approximate the linear combination of the parameters of interest, giving a combined emissions total $\eta*$, using equations 12 and 13 and transforming the predicted latent field by the weightings containing the area information.

Please also note the supplement to this comment: https://www.geosci-model-dev-discuss.net/gmd-2019-66/gmd-2019-66-AC1-supplement.pdf

―――――――――――――――――

[Figure]

**Supplement:**

[revised manuscript text omitted]

---

## Author Response (AR2)

We would again like to thank the reviewers and the editor for their comments on our discussion article 'Bayesian spatiotemporal inference of trace gas emissions using an integrated nested Laplacian approximation and Gaussian Markov random fields' (gmd-2019-66). We have submitted the revised manuscript with revisions indicated in red. Below we address each comment in turn, where the comment has been italicised.

**Reviewer 1**

*Sec. 2.5: please define the terms in your equations. For instance, what are the subscript indices -j and i in (12) and (13)? In (14), what does the right-hand side mean? You may be using a notation convention here, but it is not necessarily well-known to readers in the inversion community. Although you provide references for further details on the method, it is still important that all terms are defined and explained in this section.*
All terms are now defined as indicated below:
(P 8, L14-15) 'The speed in this approach comes from solving the marginal posteriors for $x_i$, i.e. each element $i$ of the latent field, through the numerical integration.'
(P8, L19) '...where $j$ is the $j$th element in $\boldsymbol{\theta}$ and $-j$ indicated all by the element $j$.'
(P8, L22-23) '...and the right hand is evaluated at $\mathbf{x}^*(\boldsymbol{\theta})$ which is the modal probability of $\mathbf{x}$ for a given $\boldsymbol{\theta}$.'

*P6, line 5-6: although the description of the method in this section has been improved, the authors did not really address my remark, that is, the sentence "We impose temporal correlation structure between measurements to be an autoregressive model of order one, where: ", followed by an equation describing the error model for the emissions x, is at the very least confusing. Please correct this sentence and clarify.*
This was a mistake in the text. This now reads (P6, L6) 'We impose the temporal correlation structure between emissions over time to be an autoregressive model of order one ...'

**Reviewer 2**

*Page 8, line 14: there is typo in the equation number, should say "Equations 12 and 13 ..."*
This has been corrected: (P8, L19) 'Equations 12 and 13 ...'

*Page 13, line 1: It should read ?Figure 4 shows ...?*
This has been corrected: (P13, L1) 'Figure 4 shows ...'

**Editorial comments**

*What is "a stochastic partial differential equation approach"? Clarify which stochastic partial differential equation and hint its role. You say further down: A stationary Gaussian random field with a Matérn covariance is the solution to a particular stochastic partial differential equation. You could write something in the lines of: "We use an integrated nested Laplacian approximation (INLA) for the Bayesian inference. The spatial correlation structure with spatial Markov properties results from the Gaussian random field being a solution a particular stochastic partial differential equation. "*

Thank you for this suggestion. We have made the change: (P2, L18) 'We use an integrated nested Laplacian approximation (INLA) for the Bayesian inference. The spatial correlation structure with spatial Markov properties results from the Gaussian random field being a solution a particular stochastic partial differential equation.'

*P3 L1 Clarify: e.g. "inferred latent parameter (the fluxes)"*
Added: (P3, L2) '...(the emission fluxes and boundary conditions) ...'

*P3 L28 Why well suited? Which are the desirable properties of the Matérn covariance function used here?*
Added: (P3, L29) '...due to its flexibility, with other correlation structures (e.g. exponential) being special cases of the Matérn family (Guttorp and Gneiting, 2006) ...'

*P4 L2 What kind of process is W(s) (eg a Wiener process?)*
Added: (P4 L2): '...$\mathcal{W}(\mathbf{s})$ is a Gaussian stochastic process ...'

*L 9 How are the \psis defines calculated? What properties do they have?*
Added: 'where $\psi_k(\cdot)$ is a piecewise linear function that in each triangle of a mesh construction of the spatial domain, where $\psi_k(\cdot)$ is 1 at vertex $k$ and 0 at all other vertices. This mesh is created using constrained refined Delaunay triangulation (Shewchuk, 2002), placing nodes at the main points of interests and infilling the rest of the space using some condition of minimum and maximum length of the vertices.'

*P7,8 Sections 2.4 and 2.5 seem somehow disconnected. Please make the links more explicit.*
The following has been added as a linking paragraph at the end of section 2.4 (P8, L6): 'A Bayesian hierarchical model requires a method of inference to estimate the parameters of interest and any parameters that are not of direct interest but required, and uncertain, in order to infer the other parameters in the hierarchy. Many methods of inference exist and have been applied to the problem of estimating emissions of trace gases (see references in section 1). A promising approach using the correlation structure in section 2.2 and 2.3 is INLA (Lindgren and Rue, 2015). Section 2.5 outlines this approach to inference.'

*P8, L14: Equations 12 and 13*
This has been corrected: (P8, L19) 'Equations 12 and 13 ...'

*P10 L9 Remind here the characteristics of a Matérn field.*
A prompt to the description of a Matérn field has been added (P11, L6):

[revised manuscript text omitted]